# A Series of Lanthanide Complexes with Keggin-Type Monolacunary Phosphotungstate: Synthesis and Structural Characterization

Vladimir S. Korenev *, Taisiya S. Sukhikh and Maxim N. Sokolov

Nikolaev Institute of Inorganic Chemistry, Siberian Branch of Russian Academy of Sciences, Lavrentieva St. 3, 630090 Novosibirsk, Russia
* Correspondence: wkorenev@niic.nsc.ru

**Abstract:** The coordination of rare-earth metal ions ($Ln^{3+}$) to polyoxometalates (POM) is regarded as a way of modifying and controlling their properties, such as single-molecular magnetism or luminescent behavior. The half-sandwich complexes of $Ln^{3+}$ with monolacunary Keggin POMs ($Ln^{3+}$/POM = 1:1) are of particular interest, since the $Ln^{3+}$ retains its ability to coordinate extra ligands. Thus, the knowledge of the exact structures of 1:1 Ln/POM complexes is important for the development of reliable synthetic protocols for hybrid complexes. In this work, we isolated three 1:1 $Gd^{3+}$/POM complexes of the general formula $Cat_4Gd(PW_{11}O_{39})\cdot xH_2O$ (Cat = $K^+$ or $Me_4N^+$). Complex $(Me_4N)_2K_2[Gd(H_2O)_2PW_{11}O_{39}]\cdot5H_2O$ (**1**) is polymeric, revealing a layered structural motif via bridging $Gd^{3+}$ and $K^+$ ions. Complexes $(Me_4N)_6K_2[Gd(H_2O)_3PW_{11}O_{39}]_2\cdot20H_2O$ (**2**) and $(Me_4N)_7K[Gd(H_2O)_3PW_{11}O_{39}]_2\cdot12H_2O$ (**3**) are classified as dimeric; the difference between them consists of the different crystal packing of the polyoxometalates, which is induced by a variation in the cationic composition. Isostructural complexes have also been characterized for praseodymium, europium, terbium and dysprosium. The coordination number of $Ln^{3+}$ (8) persists in all the compounds, while the binding mode of the POM varies, giving rise to different architectures with two or three $H_2O$ co-ligands per $Ln^{3+}$. However, whatever the particular structure and exact composition, the $\{Ln(PW_{11}O_{39})\}$ moieties are *always* involved in bonding with each other with the formation either of polymeric chains or dimeric units. In water, these aggregates can dissociate with the formation of $[Ln(H_2O)_4PW_{11}O_{39}]^{4-}$. This behavior must be taken into account when choosing L for the design of hybrid $\{Ln(L)POM\}$ complexes.

**Keywords:** lanthanide substituted; polyoxotungstate; monovacant Keggin-type

## 1. Introduction

The class of polyoxometalates (POMs) is characterized by great structural diversity with a small number of stable, well-studied structural types [1–3]. Heterometallic derivatives of POMs obtained for a large number of chemical elements with different electronic configurations exhibit various electronic [4,5], magnetic [6–10], catalytic [11–14], optical [15], or other useful properties [16,17]. The wide possibilities of modification determine POM applicability in such areas as biology and medicine, electrochemistry, materials and surface sciences, solar energy conversion, etc. [18–21]. It has been shown that almost all transition and non-transition metals can be included in the vacancy of lacunary Keggin-type POMs, which are the most common [22].

The combination of polyoxometalates with lanthanide (Ln) ions is also interesting from the point of view of unique functional properties, such as, for example, luminescence [23]. One of the principles for the formation of new functional materials is the binding of POM fragments through the Ln ions due to their large coordination numbers, resulting in solid-state oligomers [24,25] and large ring structures [26,27]. POMs can serve as connectors and transfer agents for different monolacunary POMs [28].

A lot of different lanthanide complexes with the monovacant Keggin-type anion, $[PW_{11}O_{39}]^{7-}$, have been well studied to date. These Ln-POMs can be divided into several structurally determined types. First of all, there are the monomeric species containing only one polyoxometalate unit. A series of chromophore-Ln-POMs $[N(CH_3)_4]_3K_2[Ln(C_7H_5O_2)(H_2O)_2(PW_{11}O_{39})]\cdot11H_2O$ (Ln = Eu, Tb, Tm, Lu) was prepared using $LnCl_3\cdot6H_2O$ and benzoic acid. Also, with the variation of the molar ratio of Eu:Tb:Tm ions, a series of multicenter-Ln analogues $[N(CH_3)_4]_3K_2[Eu_xTb_yTm_{1-x-y}(C_7H_5O_2)(H_2O)_2(PW_{11}O_{39})]\cdot11H_2O$ were synthesized [29]. More organic–inorganic hybrid lanthanide-based POMs, $[N(CH_3)_4]_3K_2[Ln(C_7H_5O_2)(H_2O)_2(PW_{11}O_{39})]\cdot11H_2O$, were synthesized using the same conditions with $HoCl_3\cdot6H_2O$ [30], $SmCl_3\cdot6H_2O$ [31] and $TbCl_3\cdot6H2O$ [32]. In all these cases, the dilacunary $[P_2W_{19}O_{69}(H_2O)]^{14-}$ was used as a source of the resulting Keggin anion. The interaction of phthalocyanine complexes of lanthanides, $[Ln(Pc)(OAc)]$ (Ln = Y, Dy, Tb), with the $[PW_{11}O_{39}]^{7-}$ anion in a 1:1:1 mixture of $CH_3CN:MeOH:CH_2Cl_2$ also yielded a row of monomeric complexes $(N(nBu)_4)_4H_2[Ln(Pc)(PW_{11}O_{39})]$ [33].

The second group includes dimeric complexes connected through the organic bridging ligands. In 2009, the series of polyoxoanions, $[\{(PW_{11}O_{39})Ln(H_2O)(CH_3COO)\}_2]^{10-}$ (Ln = Sm, Eu, Gd, Tb, Ho, Er), constructed from two monolanthanide substituted $\{PW_{11}\text{-Ln}\}$ units bridged by two acetato ligands, was published [34]. The row was continued with Dy-, Y- and Lu-containing analogues in 2012 [35]. In 2013, $Na_2[Cu(en)_2(H_2O)]_4[\{(PW_{11}O_{39})Sm(H_2O)(CH_3COO)\}_2]\cdot10H_2O$ [36] and $K_2[Cu(en)_2(H_2O)]_4[\{(PW_{11}O_{39})Tb(H_2O)(CH_3COO)\}_2]\cdot15H_2O$ [37] had the same hybrid acetate-bridging dimeric structures modified by $[Cu(en)_2(H_2O)]^{2+}$ (en = 1,2-ethylenediamine) fragments. In 2012, four new oxalate-bridging lanthanide-substituted phosphotungstates $\{[(PW_{11}O_{39})Ln(H_2O)]_2(C_2O_4)\}^{10-}$ (Ln = Y, Dy, Ho, Er) were synthesized by the reaction of a Keggin-type anion with Ln cations and oxalate ligands in aqueous solution [38]. The double-tartaric bridged phosphotungstates formulated as $[N(CH_3)_4]_6K_3H_7[Ln(C_4H_2O_6)(PW_{11}O_{39})]_2\cdot27H_2O$ (Ln = Dy, Er, Yb [39] and Tm [40]) have been obtained using the dilacunary $[P_2W_{19}O_{69}(H_2O)]^{14-}$ as a precursor. In 2012, the hydrothermal conditions were successfully used to prepare the three-dimensional architectures $[Ln(HL)(L)(H_2O)_6\{Ln(H_2L)_{0.5}(PW_{11}O_{39}H)Ln(H_2O)_4\}]_2\cdot nH_2O$ (Ln = La, Ce, Pr; $H_2L$ = 2,5-pyridinedicarboxylic acid) based on lanthanide-substituted POM units [41].

The largest number of examples to date refers to the third type—dimeric complexes in which POM fragments are linked directly through lanthanide cations. In 1971, Peacock and Weakley isolated several dimeric compounds, formulated as $K_{11}[Ce^{III}(PW_{11}O_{39})_2]\cdot24H_2O$, $K_{10}[Ce^{III}(PW_{11}O_{39})_2]\cdot25H_2O$, $K_{11}[Pr(PW_{11}O_{39})_2]\cdot22H_2O$ $K_{11}[Nd(PW_{11}O_{39})_2]\cdot26H_2O$ [42]. In 2004, the structure of $Cs_{11}Eu(PW_{11}O_{39})_2]\cdot28H_2O$ was published [43]. The $Ce^{IV}$ sandwiched between two lacunary Keggin-type $[PW_{11}O_{39}]^{7-}$ anions resulted in a lanthanide polyoxometalate $(NH_4)_2[N(CH_3)_4]_6Na_2[Ce(PW_{11}O_{39})_2]\cdot14H_2O$ [44]. The central $\{Ce^{IV}O_8\}$ polyhedron reveals the square-antiprismatic geometry. The analogues dimeric compound was further prepared for praseodymium, $[(CH_3)_4NH]_4Na_3[Pr(PW_{11}O_{39})_2]\cdot12H_2O$ [45]. The two isostructural anions, $[Ce^{III}(PW_{11}O_{39})_2]^{11-}$ and $[Ce^{IV}(PW_{11}O_{39})_2]^{10-}$, were isolated as dimethylammonium salts with the same square-antiprismatic coordination of the central $\{Ce^{III/IV}O_8\}$ polyhedron [46]. Interestingly that for similar complexes, $[N(CH_3)_4]_{10}H[La(PW_{11}O_{39})_2]\cdot21H_2O$ and $[N(CH_3)_4]_{10}H[Ce(PW_{11}O_{39})_2]\cdot19H_2O$, an approximate cubic $\{LnO_8\}$ geometry was found [47]. Potassium- or mixed potassium/caesium salts of $[Ln(PW_{11}O_{39})_2]^{11-}$ dimeric anions were obtained and characterized for ten lanthanids (Ln = Pr, Nd, Eu, Gd, Tb, Dy, Ho, Er, Tm and Yb) [48]. Three phosphors based on polyoxotungstates, $K_3Cs_8[Eu(PW_{11}O_{39})_2]\cdot11H_2O$, $K_3Cs_8[Sm(PW_{11}O_{39})_2]\cdot10H_2O$ and $K_5Cs_6[Dy(PW_{11}O_{39})_2]\cdot15H_2O$, were prepared and characterized [49]. In 2017, a dysprosium-containing phosphotungstate compound, $K_2[N(CH_3)_4]_5H_4[Dy(PW_{11}O_{39})_2]\cdot21H_2O$, was published [50].

Separately, a series of novel polyoxometalate trimers, $H_3[N(CH_3)_4]_{14}[NaLn(PW_{11}O_{39})_3]\cdot18H_2O$ (Ln = Nd, Sm, Eu), can be noted. Herein, the dimeric unit $\{Ln(PW_{11}O_{39})_2\}$ and Na-substituted POM monomer $\{NaPW_{11}O_{39}\}$ linked to form an unprecedented linear structure [51].

Organic–inorganic hybrid enantiomeric compounds of $K_{1.3}Na_{3.2}H_{6.5}[L\text{-}Pr(PW_{11}O_{39})_2]\cdot$ $8.3L\text{-}proline\cdot21.5H_2O$, $K_{1.3}Na_{3.2}H_{6.5}[D\text{-}Pr(PW_{11}O_{39})_2]\cdot8.3D\text{-}proline\cdot17H_2O$, $K_{1.3}Na_{3.2}H_{6.5}$ $[L\text{-}Er(PW_{11}O_{39})_2]\cdot8.3L\text{-}proline\cdot22.5H_2O$ [52] and $KNa_3[Hproline]_7[Sm(PW_{11}O_{39})_2]\cdot D\text{-}proline\cdot$ $18H_2O$ [53] were obtained by using L- and D-proline as chiral auxiliary agents. The series was expanded by the isolation of $[Ln(PW_{11}O_{39})_2]^{11-}$ polyanions (Ln = La, Pr, Nd, Sm, Eu, Gd, Tb, Dy, Er, Tm, Yb and Y) using proline [54].

In 2008, compound $[Cu(en)_2]_2H_8[Gd(PW_{11}O_{39})_2]\cdot(H_2en)_{0.5}\cdot3H_2O$ prepared under hydrothermal conditions was described [55]. In 2010, more inorganic–organic hybrid Ln-POM compounds with en-copper-complexes, $H_8[Cu(en)_2H_2O]_4[Cu(en)_2]\{[Cu(en)_2][La(PW_{11}O_{39})_2]\}_2\cdot$ $18H_2O$, $H_6[Na_2(en)_2(H_2O)_5][Cu(en)_2H_2O]_4[Cu(en)_2]\{[Cu(en)_2][Ce(PW_{11}O_{39})_2]\}_2\cdot16H_2O$, $H_6[Na_2(en)_2(H_2O)_5][Cu(en)_2H_2O]_4[Cu(en)_2]\{[Cu(en)_2][Pr(PW_{11}O_{39})_2]\}_2\cdot18H_2O$, $H_6[Na_2$ $(en)_2(H_2O)_4][Cu(en)_2H_2O]_4[Cu(en)_2]\{[Cu(en)_2][Nd(PW_{11}O_{39})_2]\}_2\cdot14H_2O$, $H_6[Na_2(en)_2(H_2O)_5]$ $[Cu(en)_2H_2O]_4[Cu(en)_2]\{[Cu(en)_2][Sm(PW_{11}O_{39})_2]\}_2\cdot20H_2O$ and $H_7[Cu(en)_2]_2[Sm(PW_{11}$ $O_{39})_2]\cdot10H_2O$, were published [56]. In 2011, complexes $H_{14}[Cu(en)_2]_4[Ce(PW_{11}O_{39})_2]_2\cdot2en\cdot$ $21H_2O$, $H_{14}[Cu(en)_2]_4[Pr(PW_{11}O_{39})_2]_2\cdot11H_2O$ and $H_8[Cu(en)_2H_2O]_4[Cu(en)_2]\{[Cu(en)_2]$ $[Pr(PW_{11}O_{39})_2]\}_2\cdot2en\cdot12H_2O$ were presented [57]. A series of 3d–4f heterometallic monovacant Keggin phosphotungstate derivatives, $[Cu(dap)(H_2O)_2]_{0.5}[Cu(dap)_2]_4H_2[Pr(PW_{11}O_{39})_2]\cdot$ $3H_2O$ (dap = 1,2-diaminopropane), $[Cu(en)_2(H_2O)]_2[Cu(en)_2]_{1.5}H_4[Pr(PW_{11}O_{39})_2]\cdot10H_2O$ [58], $[Cu(en)_2]_2H_6[Ce(PW_{11}O_{39})_2]\cdot8H_2O$, $[Cu(dap)_2(H_2O)][Cu(dap)_2]_{4.5}[Dy(PW_{11}O_{39})_2]\cdot4H_2O$ [59], $[Cu(dap)_2]_{5.5}[Y(PW_{11}O_{39})_2]\cdot4H_2O$ [60], $Cu(dap)_2(H_2O)][Cu(dap)_2]_{4.5}[Sm(PW_{11}O_{39})_2]\cdot5H_2O$, $[Cu(dap)_2(H_2O)][Cu(dap)_2]_{4.5}[Er(PW_{11}O_{39})_2]\cdot4H_2O$ [61], $[Cu(dap)_2(H_2O)][Cu(dap)_2]_{4.5}[La$ $(PW_{11}O_{39})_2]\cdot4H_2O$ [62] and $H[Cu(dap)_2(H_2O)][Cu(dap)_2]_4[Eu(PW_{11}O_{39})_2]\cdot13H_2O$ [63], was prepared in hydrothermal conditions.

The first 2:2 type monolanthanide substituted POM dimers, $[\{(PW_{11}O_{39}H)Ln(H_2O)_3\}_2]^{6-}$ (Ln = Nd, Gd), have been published in 2009. The binding of the $[PW_{11}O_{39}Ln(H_2O)_3]^{4-}$ anionic fragments in this structure was provided by two Ln-O-W bridges [34]. In 2017, a dysprosium-containing 2:2 type dimer, $Na_2[N(CH_3)_4]_4H_2[\{Dy(PW_{11}O_{39})(H_2O)_3\}_2]\cdot28H_2O$, was obtained [50].

One more types of compounds include the chains built from Ln-POMs. Complex $Al(H_3O)[Eu(H_2O)_2PW_{11}O_{39}]\cdot20H_2O$ was obtained in 2004, and in the solid state, it was an infinite one-dimensional zigzag polymer. A Europium atom coordinated to the POM vacant site was served as the linking element and connected to neighboring $\{PW_{11}O_{39}\}$ units via terminal oxygen atoms [43]. In 2011, the same 1-D zigzag chain was found for $[N(CH_3)_4]_4[Ce(H_2O)_2(PW_{11}O_{39})]\cdot2H_2O$ with linking $[Ce(H_2O)_2]^{3+}$ cations [64]. Another chain-like structure was found for $[N(CH_3)_4]_4[Tb(H_2O)_2(PW_{11}O_{39})]\cdot2H_2O$. In this case, the $Tb^{3+}$ cation coordinated with the POM unit vacancy was connected to the terminal oxygen of only one adjacent $\{PW_{11}O_{39}\}$ to form a linear chain [65].

Herein, we present a series of 1:1 $Ln^{3+}$/POM complexes (Ln = Pr, Eu, Gd, Tb, Dy) of the general formula $Cat_4Ln(PW_{11}O_{39})\cdot xH_2O$ (Cat = $K^+$ or $Me_4N^+$). In the case of gadolinium compounds, we found three types of structures: complex $(Me_4N)_2K_2[Gd(H_2O)_2PW_{11}O_{39}]\cdot$ $5H_2O$ (**1Gd**) is polymeric, revealing a layered structural motif, and complexes $(Me_4N)_6K_2$ $[Gd(H_2O)_3PW_{11}O_{39}]_2\cdot20H_2O$ (**2Gd**) and $(Me_4N)_7K[Gd(H_2O)_3PW_{11}O_{39}]_2\cdot12H_2O$ (**3Gd**) are classified as dimeric. For the sake of simplicity, we formulate the compounds with the $K^+$ ions outside the brackets, even though they coordinate directly with the POMs. Other Ln-containing complexes are isostructural with the gadolinium analogues.

## 2. Results and Discussion

### 2.1. Synthesis

Several approaches were described in the literature for the preparation of Ln-POM complexes differing in the lacunary phosphotungstate precursors, such as $[PW_9O_{34}]^{9-}$ [43] or $[P_2W_{19}O_{69}(H_2O)]^{14-}$ [50]. In our work, we used a monolacunary $[PW_{11}O_{39}]^{7-}$ polyoxoanion to react with Ln cations, and the pH value was set at 5.4. As a result, we obtained crystals of three different kinds simultaneously. We were succeeded in structurally characterizing all three types of crystals for the gadolinium-containing compounds, and the

unit cell parameters for the three types were also determined in the case of europium. For praseodymium, it was possible to determine the parameters of two types of crystals, and only one type was characterized in the case of dysprosium and terbium (Table 1). The resulting phases were additionally characterized by IR spectroscopy (Figure S1).

**Table 1.** Unit cell parameters for selected crystals of the compounds: V ($\text{Å}^3$), a ($\text{Å}$), b ($\text{Å}$), c ($\text{Å}$), $\alpha$ ($°$), $\beta$ ($°$), and $\gamma$ ($°$).

| Ln | 1, *P*–1 | | 2, *P*–1 | | 3, *P2$_1$/c* | |
|---|---|---|---|---|---|---|
| Pr | 2498 | 11.82 12.28 19.62 94.6 91.0 118.1 | | | 6053 | 13.11 22.08 21.35 90 101.7 90 |
| Eu | 2386 | 11.8 11.9 19.4 96.0 92.1 117.7 | 3305 | 12.81 13.04 23.41 73.1 83.9 62.1 | 5856 | 13.0 22.1 20.7 90 100.0 90 |
| Gd | 2380 | 11.73 11.92 19.39 95.85 92.30 117.57 | 3289 | 12.8 13.0 23.3 73.8 84.3 61.8 | 5799 | 12.96 22.10 20.68 90 100.11 90 |
| Tb | 2407 | 11.75 11.97 19.53 96.1 92.1 117.7 | | | | |
| Dy | | | 3267 | 12.72 13.04 23.20 74.0 81.3 62.1 | | |

### 2.2. Structural Description

The structure of $(Me_4N)_2K_2[Gd(H_2O)_2PW_{11}O_{39}]\cdot 5H_2O$ (**1Gd**) is presented in Figure 1a,b. The Gd is incorporated to the vacant site of the $\{PW_{11}O_{39}\}$ POM, coordinating via four O atoms (Figure 1a). The Gd further coordinates two neighboring POMs, thus forming a chain with the $\{Gd_2(POM)_2\}$ secondary building unit (Figure 1b), similarly to that in works [43,64]. The coordination sphere of Gd is supplemented by two $H_2O$ ligands, resulting in a $\{GdO_8\}$ polyhedron with the typical Gd–O distances. Potassium counterions join the chains into layers via K–O bonds with the POMs (Figure 2a,b). $Me_4N^+$, and disordered hydrate molecules are located between the layers.

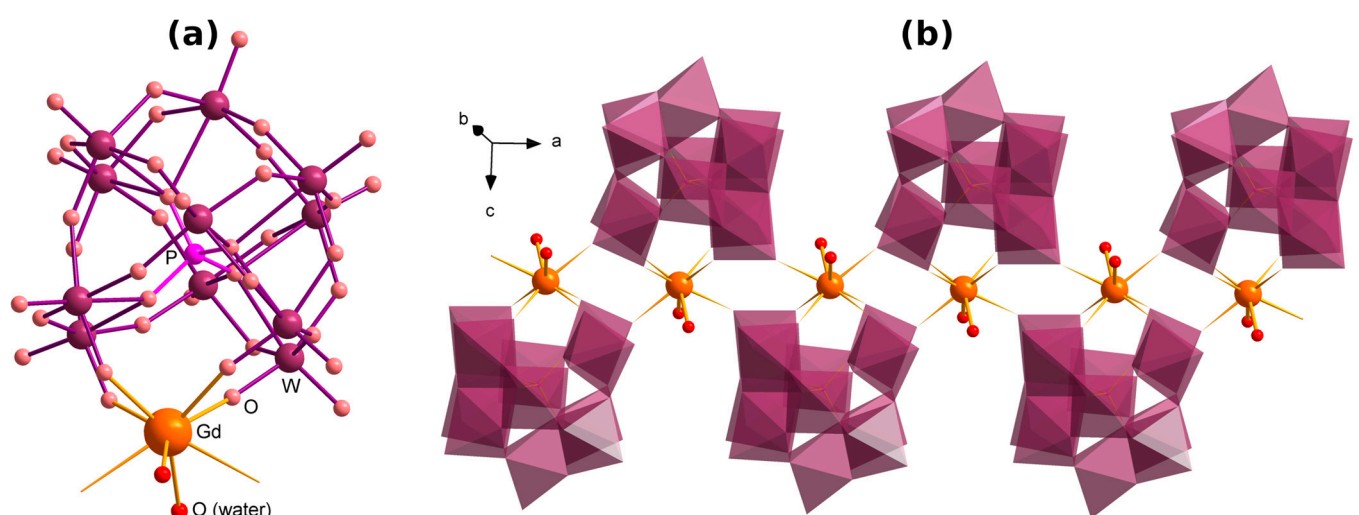

**Figure 1.** (**a**) The structure of $\{Gd(H_2O)_2PW_{11}O_{39}\}$ unit in compound **1Gd**; (**b**) the corresponding $\{Gd(H_2O)_2PW_{11}O_{39}\}_n$ chain. The $WO_6$ moieties are depicted as octahedra.

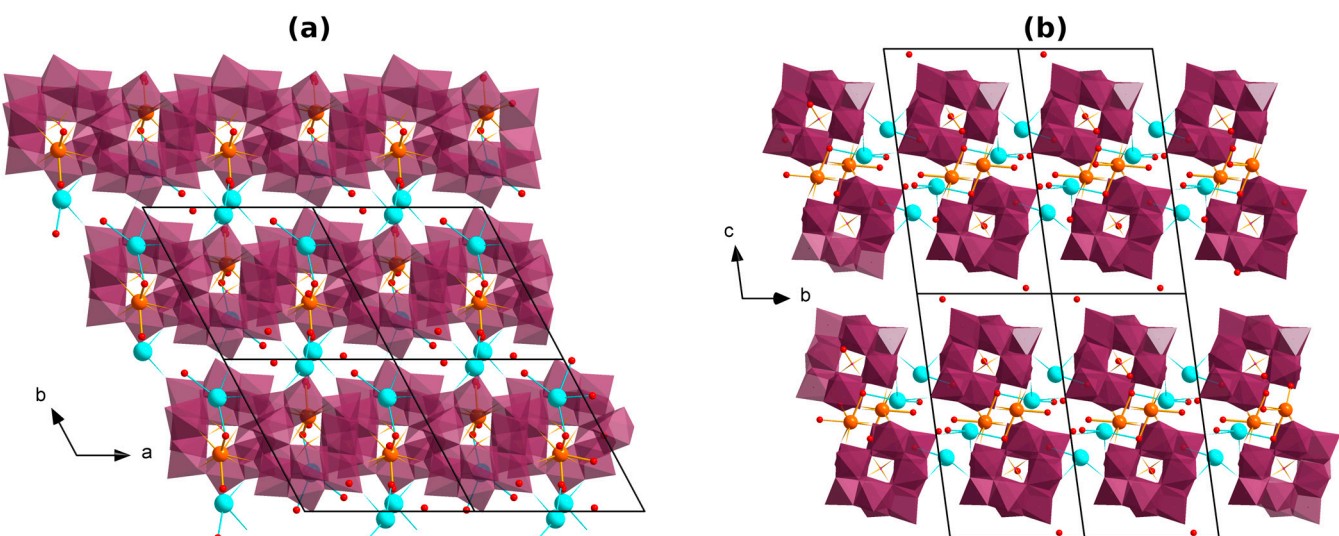

**Figure 2.** (**a**) The structure of a layer composed from $\{Gd(H_2O)_2PW_{11}O_{39}\}_n$ chains bound with $K^+$ cations in compound **1Gd**; (**b**) the crystal packing of the layers. $Me_4N^+$ and hydrogens are not shown, the $WO_6$ moieties are depicted as octahedra.

The structure of **2Gd** and **3Gd** somewhat resembles **1Gd** in that they comprise a $\{Gd_2(POM)_2\}$ secondary building unit. However, in **2Gd** and **3Gd**, it is not interlocked in a chain but is supplemented by two terminating $H_2O$ ligands, thus forming a centrosymmetric dimer structure (Figure 3). Analogous dimeric structures with different counterions ($Na^+/H^+/Me_4N^+$) have recently been reported [34,50]. The coordination polyhedron $\{GdO_8\}$ is similar in the compounds. Contrary to **1Gd**, the structures of **2Gd** and **3Gd** reveal a disorder of $K^+$ and $Me_4N^+$, i.e., the latter take partial occupancy, sharing similar positions with the hydrate molecules. The ratio of the cations is approximately similar within the series (tested for several crystals of **2** or **3**). Due to the complicated disorder, positions of the counterions and the hydrate molecules are rather arbitrary. We speculate that the disordered $K^+$ can both coordinate directly with the POM and bind via a hydrogen bond network. The number of the outer-sphere $H_2O$ and $Me_4N^+$ in **2Gd** and **3Gd** is larger than in **1Gd**, so the POM species are more separated from each other in the former. The difference between **2Gd** and **3Gd** consists in different crystal packing of the POMs, which is induced by a variation in the outer-sphere composition (Figure 4). Specifically, in **2Gd**, the POMs are located one above the other with the translation relation (*P*–1 space group), while in **3Gd**, neighboring POMs are oriented by an angle of ca. 40° with respect to each other (relation by two-fold screw symmetry; *P*2₁/*c* space group).

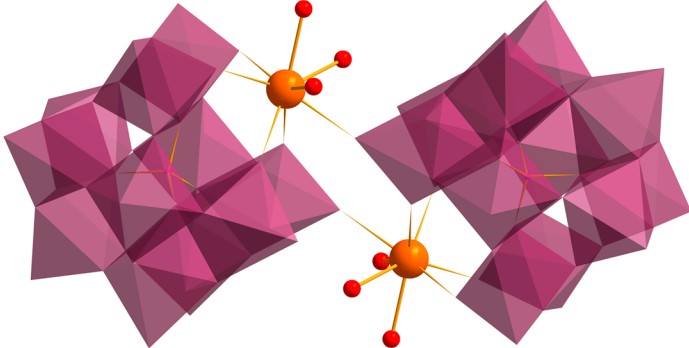

**Figure 3.** The structure of a dimer $\{Gd(H_2O)_3PW_{11}O_{39}\}_2$ in compounds **2Gd** and **3Gd**.

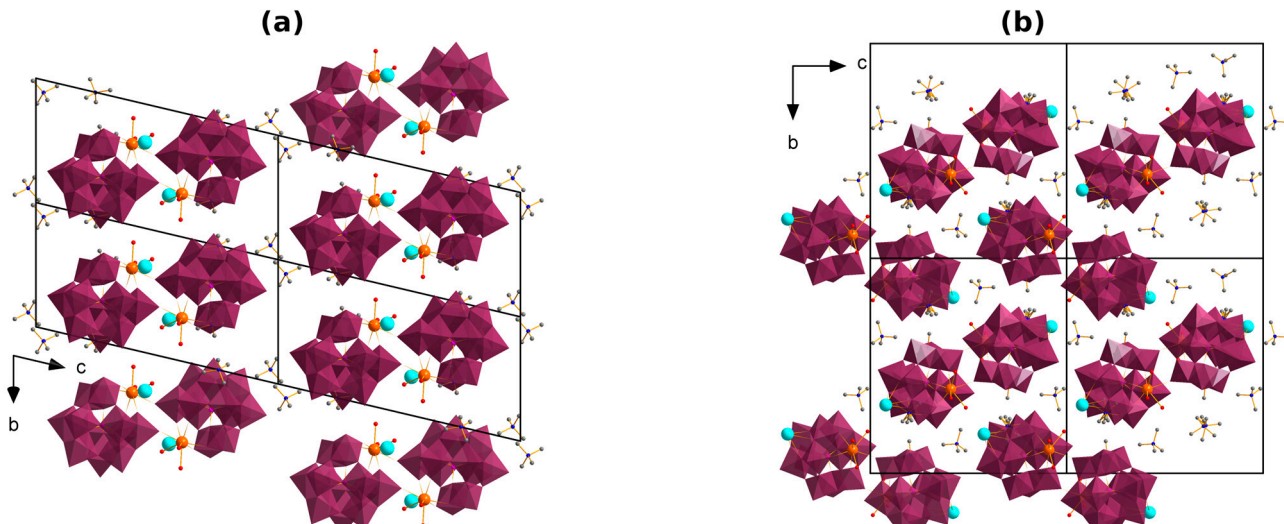

**Figure 4.** Crystal packing of the dimers in compounds **2Gd** (**a**) and **3Gd** (**b**). The $WO_6$ moieties are depicted as octahedra, hydrate molecules, hydrogens and the disorder are not shown.

Other lanthanide analogues with Pr, Eu, Tb and Dy also constitute three structural types **1–3**, which are identified by unit cell parameters of single crystals (Table 1). One can note that the unit cell parameters generally correlate with the ionic radius of the lanthanides. At the same time, experiments within one lanthanide series reveal a quite large variation of the parameters above the instrument accuracy. Specifically, the unit cell volume for several crystals of **3Pr** varies by 100 Å$^3$, while the difference in parameter $c$ can be up to 0.46 Å (Table 2). This is likely a consequence of the non-stoichiometric composition of disordered hydrate molecules in the compounds, which varies from crystal to crystal.

**Table 2.** Unit cell parameters for several crystals of **3Pr** ($P2_1/c$ space group): V (Å$^3$), a (Å), b (Å), c (Å), and β (°).

|           | *V*         | *a*           | *b*           | *c*           | β            |
|-----------|-------------|---------------|---------------|---------------|--------------|
| crystal 1 | 5969.5 (9)  | 13.0360 (8)   | 21.9985 (11)  | 21.228 (2)    | 101.300 (9)  |
| crystal 2 | 5889.3 (9)  | 13.0069 (10)  | 22.0055 (16)  | 20.908 (3)    | 100.228 (9)  |
| crystal 3 | 5878.9 (6)  | 13.0061 (7)   | 22.0170 (14)  | 20.858 (2)    | 100.174 (6)  |
| crystal 4 | 5851.2 (5)  | 13.0038 (10)  | 21.9926 (12)  | 20.7695 (12)  | 99.905 (5)   |

## 3. Experimental Section

### 3.1. Materials and Methods

The $K_7[PW_{11}O_{39}]\cdot14H_2O$ precursor was prepared according to the literature [66] and confirmed by IR spectra. Other chemicals were purchased commercially from Sigma Aldrich and used without further purification. Infrared spectra (4000–400 cm$^{-1}$) were recorded on a Scimitar FTS 2000 spectrophotometer in KBr pressed pellets.

### 3.2. Synthesis

**Synthesis of $Cat_4Gd(PW_{11}O_{39})\cdot xH_2O$:** First, 1.00 g (0.31 mmol) of $K_7[PW_{11}O_{39}]\cdot14H_2O$ was dissolved in 15 mL of water, which was followed by the addition of 0.20 g (0.54 mmol) of $GdCl_3\cdot6H_2O$ in 15 mL of water dropwise, resulting in a pH = 4.7. The pH value was adjusted to 5.4 using 2M KOH solution under stirring. The reaction mixture was heated at 80 °C for 1 h after the solution was cooled to room temperature and filtered. Then, 0.20 g (1.30 mmol) of tetramethylammonium bromide was added under stirring. After 0.5 h, the resulting solution was filtered and left to evaporate at room temperature. A crystalline phase was obtained after one week. Yield: ca. 43% (based on $K_7[PW_{11}O_{39}]\cdot14H_2O$).

**Synthesis of Cat$_4$Pr(PW$_{11}$O$_{39}$)·xH$_2$O:** The synthetic procedure was identical to that for Gd-POMs, but we used 0.2 g (0.54 mmol) of PrCl$_3$·7H$_2$O as the Ln reagent. A crystalline phase was obtained after one week. Yield: ca. 35% (based on K$_7$[PW$_{11}$O$_{39}$]·14H$_2$O).

**Synthesis of Cat$_4$Eu(PW$_{11}$O$_{39}$)·xH$_2$O:** The synthetic procedure was identical to that for Gd-POMs, but we used 0.2 g (0.55 mmol) of EuCl$_3$·6H$_2$O as the Ln reagent. A crystalline phase was obtained after one week. Yield: ca. 51% (based on K$_7$[PW$_{11}$O$_{39}$]·14H$_2$O).

**Synthesis of Cat$_4$Tb(PW$_{11}$O$_{39}$)·xH$_2$O:** The synthetic procedure was identical to that for Gd-POMs, but we used 0.2 g (0.54 mmol) of TbCl$_3$·6H$_2$O as the Ln reagent. A crystalline phase was obtained after one week. Yield: ca. 24% (based on K$_7$[PW$_{11}$O$_{39}$]·14H$_2$O).

**Synthesis of Cat$_4$Dy(PW$_{11}$O$_{39}$)·xH$_2$O:** The synthetic procedure was identical to that for Gd-POMs, but we used 0.2 g (0.53 mmol) of DyCl$_3$·6H$_2$O as the Ln reagent. A crystalline phase was obtained after one week. Yield: ca. 27% (based on K$_7$[PW$_{11}$O$_{39}$]·14H$_2$O).

### 3.3. X-Ray Diffraction on Single Crystals

Single-crystal XRD data for the compounds were collected at 150 K (Table S1) with a Bruker D8 Venture diffractometer with a CMOS PHOTON III detector (Bruker, Madison, Wisconsin, USA) and IμS 3.0 microfocus source (MoK$_\alpha$ radiation ($\lambda$ = 0.71073 Å), collimating Montel mirrors; Incoatec GmbH, Geesthacht, Germany). The crystal structures were solved using the SHELXT [67] and were refined using the SHELXL [68] programs with OLEX2 GUI [69]. Atomic displacement parameters for non-hydrogen atoms were refined in anisotropic approximation with the exception for the disordered hydrate molecules and Me$_4$N$^+$. For the latter, EADP constraints and SADI restraints were applied where needed. Hydrogen atoms were placed geometrically and refined in the riding model with the exception for those of the disordered hydrate molecules, which were not localized. The structures of **1–3Gd** were deposited to the Cambridge Crystallographic Data Centre (CCDC) as a supplementary publication, No. 2270528-2270530.

### 4. Conclusions

A series of 1:1 Ln$^{3+}$/POM (Ln = Pr, Eu, Gd, Tb, Dy) complexes of the general formula Cat$_4$Ln(PW$_{11}$O$_{39}$)·xH$_2$O (Cat = K$^+$ or Me$_4$N$^+$) was synthesized in an aqueous solution. It was shown that the {Ln(PW$_{11}$O$_{39}$)} fragments are always involved in binding with each other to form either polymer chains or dimeric units, and the Ln$^{3+}$ coordination number is the same in all cases and equals 8. It was also found that for the isostructural compounds, the unit cell parameters mainly correlate with the corresponding ionic radius of the lanthanides. The description of these compounds is important for the study of other monovacant POMs of the Keggin type in an aqueous solution and should be taken into account when developing {Ln-POM} hybrid complexes.

**Supplementary Materials:** The supporting information can be downloaded at: https://www.mdpi.com/article/10.3390/inorganics11080327/s1, Figure S1: IR spectra of Cat$_4$Ln(PW$_{11}$O$_{39}$)·xH$_2$O complexes (Ln from the top to the bottom): Gd, Eu, Pr, Tb and Dy; Table S1. Crystal data and structure refinement for **1–3Gd**.

**Author Contributions:** Conceptualization, M.N.S.; methodology, V.S.K. and M.N.S.; analysis, V.S.K. and T.S.S.; investigation, V.S.K. and T.S.S.; writing—original draft preparation, V.S.K. and T.S.S.; writing—review and editing, V.S.K. and M.N.S.; visualization, V.S.K. and T.S.S. All authors have read and agreed to the published version of the manuscript.

**Funding:** This work was supported by the Ministry of Science and Higher Education of the Russian Federation, № 121031700313-8 (The Nikolaev Institute of Inorganic Chemistry SB RAS).

**Data Availability Statement:** The data presented in this study are available on request from the corresponding author.

**Acknowledgments:** The authors thank the NIIC SB RAS team for the access to IR and SCXRD facilities.

**Conflicts of Interest:** The authors declare no conflict of interest.

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
