# Peer review of "A Series of Lanthanide Complexes with Keggin-Type Monolacunary Phosphotungstate: Synthesis and Structural Characterization"

_inorganics, doi:10.3390/inorganics11080327_

Round 1
Reviewer 1 Report
Incorporation of lanthanide ions into POMs provides a novel strategy to design functional materials. The authors synthesized and structurally characterized a series of 1:1 Ln3+/POM complexes (Ln = Pr, Eu, Gd, Tb, Dy). Three types of structures are found for all lanthanide complexes and the coordination environment of Ln3+ ions are addressed in detail. The ionic radius of lanthanides and the non-stoichiometric composition of hydrate molecules are likely to be responsible for the variation of structural parameters. This study will be of great importance in understanding the structures of complexes with Keggin-type monolacunary POMs. Considering the important findings in this work, I suggest publication of this manuscript after modifying the following typo and error.
(1) In the title, letter “y” is missed for monolacunary.
(2) For lanthanides like Pr, Tb and Dy, only one or two structural types are presented. Are other structural types not stable or not successfully synthesized?
(3) The difference in parameter c for 3Pr can be up to 0.46 Å, not 0.45 Å.
Author Response
Thank you for your kind attention to our manuscript!
(1) In the title, letter “y” is missed for monolacunary.
Response 1: Completely agree with the comment, we made a typo.
(2) For lanthanides like Pr, Tb and Dy, only one or two structural types are presented. Are other structural types not stable or not successfully synthesized?
Response 2: The synthetic procedure resulted in the formation of a mixture of crystals of the three described types in all cases. The various crystals formed simultaneously, not sequentially. For the three indicated lanthanides, we failed to select samples suitable for X-ray diffraction analysis.
(3) The difference in parameter c for 3Pr can be up to 0.46 Å, not 0.45 Å.
Response 3: We agree with the remark, there was an error in the calculation.
Reviewer 2 Report
This paper reports the syntheses of a series of lanthanide POM based complexes with various Ln salts. Mixtures of crystals were identified in this work showing different topologies (dimers or chains).
Did the authors record Powder X-ray diffraction patterns in order to compare them to the calculated ones of the different phases? A discussion on the parameters responsible of the different types of crystals would be interesting. For example, what is the reason or hypothesis explaining why this is not for all lanthanide ions?
This work is suitable for publication in Inorganics after addressing these points.
Author Response
Thank you for your kind attention to our manuscript!
Did the authors record Powder X-ray diffraction patterns in order to compare them to the calculated ones of the different phases? A discussion on the parameters responsible of the different types of crystals would be interesting. For example, what is the reason or hypothesis explaining why this is not for all lanthanide ions?
Response 1: The synthetic procedure resulted in the formation of a mixture of crystals of the three described types in all cases. The various crystals formed simultaneously, not sequentially. For three lanthanides, we failed to find samples suitable for X-ray diffraction analysis.
The resulting samples contained several phases, so we did not perform powder diffraction analysis. We plan to continue work on the separate production of pure phases, but so far nature is winning over us.